# Displacement, Polygyny, Romantic Jealousy, and Intimate Partner Violence: A Qualitative Study among Somali Refugees in Ethiopia

**DOI:** 10.3390/ijerph19095757

**Published:** 2022-05-09

**Authors:** Farida Abudulai, Marjorie Pichon, Ana Maria Buller, Jennifer Scott, Vandana Sharma

**Affiliations:** 1Department of Global Health and Development, London School of Hygiene and Tropical Medicine, 15-17 Tavistock Place, London WC1H 9SH, UK; farida.abudulai1@alumni.lshtm.ac.uk (F.A.); ana.buller@lshtm.ac.uk (A.M.B.); 2Department of Obstetrics and Gynecology, Beth Israel Deaconess Medical Center, Boston, MA 02215, USA; jscott3@bidmc.harvard.edu; 3Harvard T.H. Chan School of Public Health, 665 Huntington Avenue, Boston, MA 022115, USA; vsharma@hsph.harvard.edu

**Keywords:** intimate partner violence, romantic jealousy, polygyny, refugees, Ethiopia, displacement, Somalia

## Abstract

Romantic jealousy is a well-known relational driver of intimate partner violence (IPV), but is under-studied among displaced and polygynous populations. This study aimed to explore factors that elicit jealousy among Somali refugees in the Bokolmayo Refugee camp in Ethiopia, and the pathways leading from jealousy to IPV against women and men, to inform interventions. We conducted an exploratory, thematic analysis of 30 in-depth interviews with both women and men who were Somali refugees, as well as elders and religious leaders, organizational and service providers, policy makers, and host community members. We found that jealousy experienced by women was elicited by an unequal distribution of money and affection between co-wives, which was exacerbated by displacement-related economic hardship, and women in monogamous partnerships suspecting their husband of having other relationships. The jealousy experienced by men was elicited by their wives’ increased financial independence and interactions with other men when working outside of the home, which became more common because of displacement-related economic hardship and relaxed patriarchal gender norms. IPV interventions should address jealousy and controlling behaviors in all relationship types. Addressing conflict and relationship dynamics in polygynous households and in humanitarian settings may require specialized content, acknowledging the complex interactions and resource allocation between co-wives. Gender-transformative interventions that move away from masculinities that are built on the provider role and the introduction of alternative masculinities could also be effective in reducing IPV in this and other similar contexts.

## 1. Introduction

Globally, it is estimated that nearly one in three (27%) women experience intimate partner violence (IPV) in their lifetime [1]. Similarly, data from the 2016 Ethiopia Demographic and Health Survey estimate that 34% of ever-married women in Ethiopia aged 15–49 years’ experience at least one type of IPV in their lifetime, while 3% of women reported perpetrating physical violence against their male partner in the past 12 months [2]. The global literature suggests that war and displacement exacerbate the drivers of IPV, and that the prevalence of IPV is higher in humanitarian settings [3,4,5]. Additionally, in sub-Saharan Africa, where IPV against women has been found to be more prevalent in polygynous relationships [6], romantic jealousy is likely a key driver of IPV [7,8]. Violence between intimate partners is driven by a wide array of factors on different levels of the socio-ecological model, including individual, relational, community, and societal factors [9]. This study focused on two under-theorized relational drivers of IPV: romantic jealousy and polygyny.

Romantic jealousy is defined as “a complex set of thoughts, feelings, and actions that follow a threat to self-esteem and/or threaten the existence or quality of a relationship. These threats are generated by the perception of a real or potential attraction between the partner and (perhaps imaginary) rival” [10]. We use the term “romantic jealousy” in this study to differentiate it from other forms of jealousy (such as between siblings or peers). Romantic jealousy can be experienced by both men and women and can—but does not have to—lead to IPV. In low- and middle-income countries, where interventions have led to increased women’s empowerment and the shifting of gender norms, there is evidence that, in some cases, these interventions can also unintentionally elicit romantic jealousy, leading to IPV against women [11,12]. This has been attributed to women having increased interactions with other men as they enter the workforce and spend more time outside of the household [11,12,13].

Polygyny, the practice of men having multiple wives, has also been linked to romantic jealousy and IPV. According to global Demographic and Health Surveys from 2000 to 2010, polygyny was reported to be legal or “generally accepted” in 33 countries around the world [14]. Polygyny has been traditionally present in cultures where the economic success of a family depends on the number of children available to work [15]. Research exploring the dynamics of polygyny have identified multiple sources of conflict between men and their wives, and among co-wives. One major source of conflict occurs when the family experiences economic hardship, and having many children in one household (as is characteristic in polygynous unions) increases the financial burden for families [15]. Other identified sources of conflict within polygynous relationships include rivalries between co-wives and, in some cases, between children of different mothers [16]. Polygyny has been widely associated with IPV [17], especially in sub-Saharan Africa [18], but little is known about the pathways through which polygyny leads to IPV. In a study with polygynous Somali refugees in Finland, participants reported romantic jealousy as the most common reason for their poor mental health [19]. In Ethiopia, a study found that the prevalence of IPV among women who were in polygynous relationships was four times higher than among women in monogamous relationships [20]. These findings suggest that romantic jealousy among polygynous Somali refugees in Ethiopia likely plays a major role in generating relationship conflicts, leading to IPV.

The lifetime prevalence of different types of IPV against men globally and in Ethiopia remain unknown. Likewise, the IPV prevalence in the Bokolmayo refugee camp in the Somali region of Ethiopia, where this study was conducted, is not known. Findings from a recent systematic review identified six pathways through which romantic jealousy experienced by men and women can lead to different forms of IPV against women (described further in the discussion) [11]. These pathways interact with factors such as a lack of communication within the couple, economic control and dependence, and male alcohol consumption to drive IPV perpetrated against women. Studies on polygynous relationships, however, were excluded from the systematic review, highlighting an important gap in the literature. Additionally, little is known about the experiences of men, both as perpetrators of IPV and as the ones experiencing IPV [21], and the pathways from romantic jealousy to IPV against men are under-researched. Finally, to our knowledge, no studies to date have explored the role of romantic jealousy in IPV among refugee populations. 

We conducted an exploratory, secondary analysis of qualitative data from men and women who are Somali refugees living in the Bokolmayo refugee camp in Ethiopia, a population in which polygyny is common, to better understand the role of romantic jealousy in IPV in this setting. This data were collected to inform the adaptation of Unite for a Better Life (UBL), an evidence-based, gender-transformative program that was effective in reducing IPV among a population in rural Ethiopia [22,23,24,25]. This study aimed to explore the factors that elicit romantic jealousy experienced by men and women in this context, and the subsequent pathways leading to IPV against both women and men, to inform interventions among refugee and polygynous populations.

## 2. Methods

Data for this secondary analysis were collected in October 2016 as part of a qualitative study conducted in collaboration with Women and Health Alliance International in Ethiopia, Addis Ababa University School of Public Health in Ethiopia, and Beth Israel Deaconess Medical Center in Boston, MA, USA. The study explored the risk factors for, and protective factors against, IPV among Somali refugees in the Bokolmayo refugee camp in Ethiopia [12,26]. The study was designed to inform the adaptation of UBL to this humanitarian context. The original intervention included 14 group-based participatory sessions on gender norms, healthy relationships, conflict resolution, and HIV, and was delivered within the context of the Ethiopian traditional coffee ceremony. The original context in which UBL was implemented included polygynous marriages, but content was not specifically tailored to address the unique risks related to polygyny. This secondary analysis was conducted June 2020–September 2021 to inform further refinements to the adapted curriculum and future interventions.

### 2.1. Study Setting

Located near Dollo Ado, a small town in southern Ethiopia that borders Somalia, Bokolmayo is one of five refugee camps in the area. At the time of the study, the United Nations High Commissioner for Refugees estimated that of the 200,000 Somali refugees that were residing in the five camps, approximately 42,385 refugees were housed at Bokolmayo, which opened in 2010 [27].

### 2.2. Data Collection

Participants were recruited by using purposive sampling to capture diverse views and perspectives on IPV and approaches to address IPV in this setting. Based on their role in the community, different inclusion criteria were applied to different sub-groups. For refugee community members, the inclusion criteria included: adolescent boys and girls, as well as men and women aged 15 years and above who were identified as Somali refugees and had resided in Bokolmayo camp for six or more months. Organizational and/or service providers had to be at least 18 years of age, a staff member or service provider from the community organizations operating in Dollo Ado, and have worked for at least one year for an agency or organization that provides services to refugee populations. Lastly, inclusion criteria for community leaders and elders included: men or women 18 years or older who were community leaders and/or religious leaders in Dollo Ado and who had been in the Bokolmayo camp for at least six months.

Thirty in-depth interviews (IDIs), ten focus group discussions (FGDs), and 13 participatory learning activities (PLAs) were conducted with community elders, community members, religious leaders, health workers, host community members, non-governmental organization (NGO) staff, and policy members in the Bokolmayo refugee camp. All interviews were conducted in either Somali or Mai-Mai, the local dialect, and then were translated and transcribed verbatim into English. Each interview was audio recorded and lasted between 40 to 90 min. Three women and seven men who were fluent in Somali and/or Mai-Mai were recruited from the refugee camp to serve as interviewers. They completed a six-day training module on qualitative methods, interviewing techniques, and protection of human subjects and risk mitigation. Interviewers worked in pairs for increased security and to ensure that there was a notetaker present. At the end of each interview, participants were given information on available medical, legal, and other relevant support services.

Semi-structured questions were developed for the IDIs and FDGs to identify risk factors that contributed to IPV. The topic guides covered marital relationships and practices before and after displacement including: (1) polygyny, (2) early and forced marriage, (3) dowry, (4) decision-making among couples, (5) conflict within marital relationships, and (6) physical and sexual IPV. There were no questions that explicitly asked about romantic jealousy, but open-ended questions on sources of relationship conflict and marital practices before and after displacement yielded responses that highlighted romantic jealousy. Examples of these questions include: “What kinds of conflicts occur in marriages and families?” and “In your opinion, do you think there are any problems in a marriage of more than one wife?”

### 2.3. Data Analysis

The FGDs and PLAs were excluded from this secondary analysis because they did not elicit data on romantic jealousy. We analyzed all IDIs, including data from staff members and service providers, because they were community member themselves, and they had worked closely with other community members and thus had a nuanced understanding of their relationship dynamics. Interviews conducted with both men and women were included for breadth and diversity of perceptions on romantic jealousy and IPV. Transcripts of the IDIs were uploaded into NVivo 12.0 qualitative analysis software. A hybrid deductive and inductive approach to Braun and Clark’s thematic analysis was used to create the codebook and code the transcripts [28]. The use of this hybrid approach allowed the analysis to include existing theories around romantic jealousy and IPV [11,13,29], while also incorporating new themes that emerged from the data.

These interviews were read once to make note of preliminary observations, as well as to determine how much information could be gathered on romantic jealousy and IPV and what data would be included in the overall analysis. With each additional appraisal of the interviews, memos with notes and observations were kept, capturing contextual information to better assist in the understanding of the socio-cultural norms in this community. The final codebook had exploratory child nodes that included differentiation between men and women’s perceptions of romantic jealousy and IPV. Examples of some of the overarching codes included “IPV elicited by romantic jealousy” and “attitudes and practices surrounding polygyny.”

We considered the following types of IPV in our analysis: (1) physical, including slapping, hitting, kicking, etc.; (2) sexual, including forced sexual intercourse or having intercourse due to fear of what the partner might do; (3) psychological, including belittling, constant humiliation, threats to take away children, etc.; (4) controlling behaviors, including isolating a person from their family and friends and monitoring their movements; (5) economic, including controlling a person’s ability to acquire, use, and maintain resources, thus, threatening their economic security and potential for self-sufficiency [30,31]. We include quotes in the results that represented most participants’ views, but also those that present diverse examples and nuanced perspectives.

### 2.4. Ethical Considerations

The original qualitative study was approved by the Institutional Review Board at Beth Israel Deaconess Medical Centre in Boston and Addis Ababa University. The United Nations High Commissioner for Refugees and the Administration for Refuge and Returnee Affairs also provided permission to conduct the research. The ethical approval for this secondary analysis was granted by the London School of Hygiene and Tropical Medicine (ref. 21945).

During the study, a local community advisory board comprising representatives from the refugee camp was regularly convened to provide input on the procedures and to ensure local oversight. On-site supervisors, interviewers, and facilitators were all required to sign a code of conduct, ensuring the protection of study participants. The code of conduct included maintaining the confidentiality and integrity of data collection, as well as fidelity to the study protocol.

Verbal consent was obtained from all participants due to an anticipated lack of literacy. Separate parental consent was not required for participants under 18 years due to potential separation from their parents during displacement, and the sensitive nature of the topic. All participants were assigned a unique identifier to ensure anonymity and privacy, and to prevent any risk associated with the sensitive nature of the interviews being revealed. Further details on the original study’s methodologies and ethical procedures can be found in Sharma and colleague’s papers [12,25,26,32].

## 3. Results

### 3.1. Participant Characteristics

Thirteen women and seventeen men (*n* = 30) participated in the IDIs. Twenty-four of the participants were originally from Somalia and the remaining six were from Ethiopia. Participant ages ranged from 17–70 years. The mean age of women was 29.1 years and that of men was 36.8 years. Twenty-two of the participants were married, seven were single, and one participant was separated. All the participants self-identified as Muslim. See Table 1 for additional participant demographic information.

### 3.2. Romantic Jealousy and IPV

We identified three pathways leading from romantic jealousy to IPV against women and men in both polygynous and monogamous relationships in the qualitative data (Figure 1). These pathways were built upon the key three themes that emerged from the analysis regarding the causes of romantic jealousy leading to IPV: (1) unequal distribution of financial resources and affection between co-wives; (2) women in monogamous relationships suspecting their partner of having another relationship; and (3) women gaining employment. Differences arose in the accounts of romantic jealousy experienced by women (Pathways 1 and 2) and men (Pathway 3). While the romantic jealousy experienced by women was cited as leading to many instances of violence, there was less discussion regarding romantic jealousy experienced by men among interviewees, and it was only linked to a few reports of IPV against women. We begin by describing women’s experiences of romantic jealousy toward their husbands, and end with a description of men’s experiences of romantic jealousy towards their wives.

#### 3.2.1. Romantic Jealousy Experienced by Women

Several of the participants reported that romantic jealousy experienced by women contributed to conflicts and stress in relationships. For example, one man stated: “Women are jealous about many things. Most of the disagreements come from women.” (24 year-old man, refugee community member, IDI 16). Two causes of romantic jealousy that were experienced by women, leading to physical IPV against women and men emerged from the data: (1) unequal distribution of financial resources and affection between co-wives in a polygynous relationship, and (2) women in monogamous relationships suspecting their husband of having another, secret relationship.

##### Unequal Treatment of Wives in Polygynous Marriages

The data suggest that marital practices among refugees in this community were different from those practiced in Somalia. Respondents reported that men and women in the refugee camp had more autonomy when choosing a spouse than prior to displacement. For example, a woman stated: “*First [before displacement] her father used to force her [to marry] without her consent […] now she comes with [marries] the man she loves*.” (19 year-old woman, refugee community member, IDI 10).

Nearly all the participants agreed that under Islamic law, men are allowed “*to marry up to four wives if he [has] the money or means for it*.” (49 year-old man, refugee community member, IDI 18). Very few participants discussed why men might choose to marry multiple wives, but one man provided the following explanation:
*Most of the time men are interested [in] a second [wife]. If he has one wife and he wants to have more children, then [he] will marry a young girl. He will be more interested [in] the [younger wife] and forget the old one*.(24 year-old man, refugee community member, IDI 16)

Only one participant offered a personal testimony for why he opted to not have a second wife for reasons beyond financial constraints. Despite being permitted by Islamic law, he described how he did not want his wife to experience the psychological distress brought by polygyny:
*We were together for 20 years and there is no other wife, because my economic [situation does] not allow me to do that. [Also] my wife, who has patiently struggled with me in hard times or [through] problems, I do not want to break her heart [or] upset [her]*.(49 year-old man, refugee community member, IDI 18)

The perception of what constitutes a good husband or wife depended on their ability to fulfill local gender norms. For men, their ability to fulfill the provider role was pivotal in judging their performance as husbands. For example, a policy maker stated: *“[The man should] make a soft [peaceful] and respect[full] relationship as well as cover the needs of the wife, whether [it is] education or service, they should cover her needs*.” (43 year-old man, policy maker, IDI 21).

Many participants across all demographic groups reported that economic hardship and the unequal treatment of co-wives were the predominant sources of relationship conflict among refugees in this community. For example, a refugee community member stated:
*He is not capable [of working] as he previously [did because of displacement]. Yes, there is a pressure [for the husband]. Even if he was able to earn [enough] money for his family, he cannot earn [enough money] after he [marries] the second wife. He is not aware of his children most of the time, the [men] are absent minded*.(24 year-old man, refugee community member, IDI 16)

Another male refugee community member agreed with the previous statement, and when asked about the causes of conflict in marriages, he explained: “*Yes sometimes [polygyny] can cause [the destruction of] the previous [first marriage], or men give the highest priority to the new wife, that leads to the neglect [of] the other wife and her children and causes problems among families.*” (29 year-old man, refugee community member, IDI 6).

Economic hardship experienced after displacement made it more difficult for multiple wives of a husband to receive equal attention and money, and participants of all age groups and positions in the community reported that many existing polygynous unions were characterized by an unequal distribution of affection and/or resources towards a man’s multiple wives. As an elder in the community reported:
*You can confirm that he has more sex [with one] than [the] other, he gives more [money to one] than the other, it was said he holds one [with his] left hand and the other [with his right hand]. For Somalis, we always give the importance to the one we have on the right hand*.(66 year-old man, clan elder, IDI 20)

The imbalance of resource provision reportedly led to competition among the co-wives. For example, a religious leader stated: “*He may align with the one who supports him [the most], or the women compete when there are too many [wives]. They compete, [and] he supports the one [who] wins in the competition*.” (51 year-old man, religious leader, IDI 14).

A health worker in the community provided an alternate insight into the relationship dynamics of polygynous marriages, stating: “*Usually the first wife, when she [has] about six children, another fresh wife follows, so the first and the second are not treated [the same]. Most men are usually [on] the side of the new wife*.” (45 year-old man, health worker, IDI 22). Another woman further highlighted the consequences for women who were perceived to have a lower position when compared to the other wife, stating:
*If he doesn’t have an income [that he can equally distribute], he will have a problem with his previous wife. [As a result] he stays with the last wife alone. He will not give anything [to the] previous wife and to not give, [is to not] take care of the children [with his first wife]*.(44 year-old woman, host community member, IDI 30)

Thus, for wives in polygynous marriages, maintaining their husband’s affection was crucial for establishing their position among other wives and for securing resources.

Confrontations precipitated by the romantic jealousy of the first wife were described by a few participants to lead to arguments and physical IPV perpetrated by men. For example, a community-based organization worker reported:
*Mostly women are jealous. This woman and her children want to be given exactly what the other woman and children are given […] the man cannot avoid that. She may shout at him and [an] argument [erupts]. After that, he gets angry and starts beating [her]*.(32 year-old woman, community-based organization worker, IDI 2)

Conflicts and physical violence between co-wives arising from romantic jealousy were also discussed by some of the participants. For example, one man stated: “*Having two wives you [are] inclined to [prefer] one wife. [The] other wife suspects, so it is possible that she may inflict [violent] acts against the other wife*.” (17 year-old man, refugee community member, IDI 11).

Violent acts that are perpetrated by women toward male partners in this situation were also described. For example, when a woman was asked for examples of violent behaviors towards men in this community, she said: “*The man [behaves badly] when he goes to [the] other wife […] As he enters the door [of the first wife’s house], she starts insulting [him and] beating [him] with a shoe*.” (19 year-old woman, refugee community member, IDI 10).

One participant also discussed how women may “*beat [her own] children*” in response to feeling neglected by her husband (30 year-old woman, refugee community member, IDI 17). This quote further exemplifies the detrimental impact that romantic jealousy can have, not only on partners, but on families and children in this setting.

##### Suspicion of Other Intimate Relationships in Monogamous Marriages

Although new polygynous marriages were no longer common occurrences in the refugee camp because of the economic hardship caused by displacement, a few participants described fearing the possibility of their husbands seeking other relationships or another wife, and explained the ways in which they tried to prevent it. For example, as one woman said about married women in the community: “*She make[s] her bed ready in the afternoon and [gets] the food ready. She has to do everything for him, so that he doesn’t look to other women*.” (19 year-old woman, refugee community member, IDI 10). Another man shared a similar opinion about women in the community, stating: “*First and foremost, women can attract her husband. If she wants to stop him from seeking a second wife, she must win his confidence*.” (29 year-old man, refugee community member, IDI 6). These quotes highlight that both men and women held the belief that to stop her husband from “*seeking a second wife*”, it was the woman’s responsibility to fulfill her marital duties, which included household tasks.

When participants were asked if a wife has the right to reject her husband’s sexual advance one participant replied: *“[They have] no right to refuse [their husband sexually], but wives [will still] reject [their husband] [laughs]*.” (46 year-old woman, refugee community member, IDI 5). Notably, one unmarried adolescent girl described the implications of a woman rejecting a husband’s request for sex, explaining: “*She refuses her husband [sex] and goes to another man*.” (16 year-old woman, refugee community member, IDI 9). In this statement, the participant highlights that not only must women complete their household duties to prevent their husband from seeking another partner, but they must also have sex with their husband as a way of verifying that they are not seeking another partner themselves.

Several participants reported that women’s suspicions of her husband entering into a sexual relationship with other women could spark romantic jealousy, which could, in turn, lead to women being physically violent towards their husbands. When asked why a woman may beat her husband, one man stated:
*She [does] so if she observes her husband with other women or if she sees him chasing after other girls […] [Women] become jealous when [they] hear that he is dating other girls or having a relationship with [other] girls. It is possible [for her] to beat [him or] even burn him alive with boiled oil*.(24 year-old man, refugee community member, IDI 16)

Other examples of women perpetrating violent acts included biting, kicking, and slapping. As a female participant described, women using physical violence against their husbands was a shift from normal behaviors women exhibited in Somalia: “*Now she can beat [her husband] but not when we were in our country. [She beats him now and not before, because she becomes] jealous [when] he marries another wife*.” (19 year-old woman, refugee community member, IDI 10). This quote suggests that while women may have experienced romantic jealousy in Somalia, they were unable to express those feelings or challenge their husbands due to restrictive gender norms; but, in the refugee camp where polygyny is practiced less often, women were more empowered to challenge their husbands. One woman discussed the potential for physical IPV and/or homicide of women or men to occur when women confronted their husbands about polygyny:
*[The woman] does not allow [her husband] to [marry another woman]. It brings beat[ings]. She may kill the man, or the man may kill the woman […] The man bringing another woman may not be allowed [by his wife] though religion does accept [it]*.(25 year-old woman, refugee community member, IDI 1)

One man suggested that in other extreme cases, romantic jealousy experienced by women could also lead to suicidal ideation: “*At the time she feels jealous. Being jealous causes her to beat her husband […] a jealous woman as she thinks too much, she could [also] endanger herself to take her own life*.” (20 year-old woman, NGO worker, IDI 24).

The examples presented in this section highlight women’s concerns about their husbands having sex with or marrying another woman. Women performing all their household duties, not refusing sex, and punishing their husbands for straying were reported as strategies they employed to maintain family dynamics, resulting in several reports of physical IPV against men and women.

#### 3.2.2. Romantic Jealousy Experienced by Men

Respondents reported that romantic jealousy experienced by men was elicited by female employment, leading to women having increased financial independence and interactions with other men. Men sometimes responded to these perceived threats with controlling behaviors, physical IPV, psychological IPV, and/or economic IPV against their wives.

Several respondents reported an increase in women entering the workforce after displacement. No consensus was reached about the acceptability of women working outside of the home, but one man reported that it could be essential for a family’s survival:
*After the collapse of [the] Somali government, the life of the family depended [on] women, as the men were mostly involved in the fighting. Then all the wealth of the families belonged to the woman or the mothers. Mostly Somalis believe that women are the backbone of the livelihood for the family*.(51 year-old man, religious leader, IDI 14)

Other members of the community described more negative feelings towards women working outside of the home. For example, one woman stated that women could not be trusted:
*She can work, but [there is] the belief that the women are vulnerable and can cheat and [keep her earning from her husband]. If they have a [job near their home], they [her family] can monitor [her] and [she can] follow up with [her duties] at home. [Then she and her husband] can both work*.(46 year-old woman, refugee community member, IDI 13)

A health worker provided more insight into why men sought to prevent their wives from working outside of the home and alluded to the use of controlling behaviors and economic IPV:
*The [men] believe that if the girl works and earns money, she will become more independent of her man and later, she will start looking for another man. A man [whose temperament] she can understand because she works, earns money and is educated. If she leaves [her] current husband and does her own business, he may be [afraid], so he will oppress her*.(45 year-old man, health worker, IDI 22)

In this quote, the participant postulates that financial independence gives women the freedom to choose if they want to stay with their husbands, and hence can be threatening for men. Thus, men may sometimes keep their wife from working outside of the home. When discussing men’s worries with respect to their wives working outside of the home, a religious leader stated that another concern was exposure to other men: “*I think the husband might suspect [that] his wife is sleeping with other [men]*.” (58 year-old man, religious leader, IDI 2). A community-based organization worker provided another example of romantic jealousy stemming from women’s exposure to other men when working outside of the home: “*He hates to see her work with others or laugh with others. [Some men] stop women [from] working due to [their] jealousy.”* (41 year-old man, community-based organization, IDI 26).

Psychological IPV and/or controlling behaviors that were perpetrated as a response to the suspicion of his wife having another relationship while outside of the household was also reported by one participant:
*There are those [who] when entering their house, begin yelling with words that could not be tolerated. His wife cannot tolerate [them]. There are those when they enter their house, they begin [to interrogate their wives] and ask questions like, ‘Where have you been? I came [by the house] before and you were not there*.(49 year-old man, refugee community member, IDI 18)

Another participant reported physical IPV perpetrated by men because of romantic jealousy. When asked why men perpetrate violence against their wives, he said: “*You may see a wife who wears short clothes and [being constantly beaten and in] severe [relationship] conflicts […] the jealousy, misunderstanding, and disobedience may bring beatings*.” (17 year-old man, refugee community member, IDI 11). This quote clearly exemplifies how unequal power dynamics within a relationship are the base of violence against women in the refugee camp.

Additionally, the evidence suggests that when men did experience romantic jealousy, women were blamed for causing the relationship conflict, as one religious leader described: “*You can see a married [woman] with another man. [Her husband] feels something […] ‘What do you want from him, you are jealous?’ she says. Most women do that, they make a problem for the man*.” (51 year-old man, religious leader, IDI 14). This final quote suggests that regardless of whether the woman experienced jealousy or her husband did, women were blamed for the subsequent relationship conflict.

## 4. Discussion

This study found that romantic jealousy, particularly romantic jealousy experienced by women, catalyzes relationship conflict and IPV against women and men in this refugee community in which polygyny is common. Three pathways in which romantic jealously contributed to relationship conflicts and IPV arose from the data (see Figure 1): (1) Romantic jealousy experienced by women in polygynous relationships and elicited by the unequal distribution of financial resources and affection between co-wives led to men perpetrating physical IPV against their wives, and women using physical violence in retaliation for men marrying other women; (2) Romantic jealousy experienced by women in monogamous relationships and elicited by the suspicion of their husband having other intimate relationships, led to accounts of confronted men perpetrating physical IPV against their wives, and women using physical violence in retaliation for men entering into relationships with other women; and (3) Romantic jealousy experienced by men was elicited by women who gained employment and led to controlling behaviors, as well as physical, psychological, and economic IPV against women.

Each pathway has been directly or indirectly influenced by changes related to displacement. In this population, displacement has led to increased poverty, a decline in polygynous marriages (although participants reported that attitudes towards polygyny have remained favorable), more women entering the workforce, and increased conflict among intimate partners, especially those in polygynous marriages. We found that the economic hardship caused by displacement led new polygynous relationships to become less common, but for those already in polygynous relationships, this exacerbated the unequal distribution of wealth between co-wives. Romantic jealousy experienced by men, on the other hand, was discussed in relation to women’s increased interactions with other men and their husband’s fear that she would no longer be financially dependent on him, and therefore would find another partner after joining the workforce. We also found that participants believed it was a wife’s responsibility to prevent her husband from seeking another relationship by completing household tasks, and that women refusing sex implied that she was in a relationship with another man. Regardless of who was jealous (the man or woman), in all three pathways romantic jealousy resulted in violence against women, while only romantic jealousy experienced by women was reported to lead to violence against men. Participants reported that violent acts against men had become more commonplace after displacement. Additionally, we found evidence of other forms of domestic violence between co-wives and against children due to romantic jealousy experienced by women, and one report of women potentially experiencing suicidal ideation because of romantic jealousy she experienced.

IPV perpetrated by women against male partners is understudied, but some evidence from several contexts suggest that it is not uncommon [33]. A review of the literature highlights that psychological IPV is the most common form of IPV perpetrated by women against men, while sexual IPV is the least common [34]. Common motivations for women perpetrating IPV include feeling as if their partner was not giving them attention, self-defense, and retaliation for violence perpetrated against them [33]. These motivations are in line with our findings, which suggest that within a patriarchal, gendered hierarchy in which men have power over their partner, violence perpetrated by men tends to be motivated by control (Pathway 3), while violence perpetrated by women tends to be motivated by retaliation (Pathways 1 and 2).

Buunk identified three types of romantic jealousy: (1) reactive jealousy, defined as a negative response resulting from the emotional or sexual involvement of a partner with someone else; (2) anxious/anticipatory jealousy, defined as resulting from the jealous partner’s fear of the possibility that their partner is in a romantic relationship with a third party; (3) preventative jealousy, defined as the jealous partner taking action to prevent the emotional or sexual contact of their partner with someone else [35]. In our study, the violence against men described was reactive and in retaliation to men taking another wife or seeking other women, whereas violence perpetrated by men included reactive, anxious, and preventative motivations. Reactive actions led to men perpetrating physical IPV against their wives when he suspected her of talking to other men, or when confronted with the accusations of favoring one wife or expressing interest in other women. In this setting, anxious romantic jealousy manifested in men’s use of controlling behaviors (preventing women from leaving the house or working) and physical IPV (beating women for wearing revealing clothing). Although women’s motivations for violence were not explored, Pichon and colleagues found a similar pattern in relation to men’s reactive, anxious, and preventative romantic jealousy [11].

Pichon and colleagues identified six pathways through which romantic jealousy and infidelity lead to IPV against women. Findings from our study support four of these pathways: (1) men who suspect their partner of infidelity use physical and psychological IPV; (2) women who suspect their partner of infidelity experience physical and psychological IPV; (3) men who anticipate partner infidelity use controlling behaviors and economic IPV; and (5) women who anticipate partner infidelity and male suspicion of their own infidelity experience sexual coercion [11]. Our study highlights that in a context of economic hardship caused by displacement, and where polygyny is common, women have an increased motivation to prevent their husband from seeking another relationship (Pichon and colleague’s Pathway 2). Additionally, Pichon and colleague’s Pathway 3 may be exacerbated by the economic hardship caused by displacement, as income brought in by women working outside of the home may become essential for the family’s survival. Our study also suggests a new pathway for polygynous relationships, in which romantic jealousy experienced by co-wives can be elicited by competition for financial resources and affection, leading to physical IPV against both women and men.

Participants described local masculinities as being defined by the provider role; with displacement, this masculinity was threatened as men were no longer able to provide for their families as well as they used to, and women increasingly joined the workforce and were relied on to provide for the family. In line with past research, we found that women entering the workforce and becoming increasingly financially autonomous also had the potential to threaten their husbands’ masculinities and lead to increased IPV against women [13,36] Among the identified risk factors for IPV, however, women’s economic dependence also features prominently in the literature [37]. This suggests that gender-transformative interventions that shift masculinities away from the provider role and providing alternative masculinities could help with this transition [37,38].

Our findings regarding the relationships among co-wives contrasts many of those in the literature. For example, Lees and colleagues [39] found that in Mali, the first wife maintained overall authority, had a closer relationship with the husband, and reported having good relationships with her co-wives. The current study did not examine the relationship between co-wives specifically, but in line with past literature, participants reported that perceived preferential treatment led to increased conflict between the first wife and/or her husband or co-wife [35]. Further research into the relationships of co-wives in the household should be considered, as it could provide insight into additional pathways by which romantic jealousy leads to IPV, as well as highlight more mechanisms which may be effective in tackling IPV in polygynous contexts.

Overall, these findings have implications for IPV prevention programs such as UBL, when they are delivered in settings where polygynous unions are prevalent. While UBL was found to be effective in reducing both the experiences and perpetration of IPV when delivered to groups of men in the overall sample [32], it is unclear as to whether there may have been differential effects among polygynous versus monogamous households. The findings of the current study suggest that tailored content to address relationship dynamics, romantic jealousy, and relationship conflict in polygynous households may be needed and could increase the impact of the intervention further [32]. Future interventions may also benefit from integrating topics about romantic jealousy and how to address it within their curricula [32].

### Limitations and Strengths

There are some limitations of this study to note. First, because this is a qualitative cross-sectional study, causality between romantic jealousy and IPV could not be determined. The aims of this study, however, were on exploring the pathways from romantic jealousy to IPV in this setting. Additionally, the original study was not designed to assess romantic jealousy, but romantic jealousy repeatedly arose during interviews, highlighting its importance within the community. Lastly, participants were not asked to disclose or discuss personal experiences of IPV to minimize the potential harms of participating in this study. The health workers and individuals working at community-based organizations, however, had experience dealing with cases of IPV, and when they spoke about IPV they were speaking about experiences that they had observed in the community. We did not find any meaningful differences across participant groups, likely because of our limited sample size.

This study also has multiple strengths. First, the study sample included various members of the refugee community (for example religious leaders and community-based organization staff members) and members from the host community. In addition, a wide range of ages were included, allowing for many different perspectives to be analyzed. Moreover, women’s and men’s perspectives were examined, which allowed for the exploration of both women’s and men’s experiences of IPV as it related to romantic jealousy.

## 5. Conclusions

This study provides evidence of the role of romantic jealousy in threatening the security of relationships and leading to experiences of IPV against women and men in polygynous and monogamous marriages. Three pathways from romantic jealousy to IPV emerged from the data (see Figure 1): in polygynous relationships, the romantic jealousy experienced by women was elicited by an unequal distribution of financial resources and affection between co-wives, while in monogamous relationships, romantic jealousy experienced by women was elicited by suspicions of their husband having other sexual relationships, both leading to men perpetrating physical IPV against their wives, and women using physical violence against their husbands. Romantic jealousy that was experienced by men was elicited by women joining the workforce and thus becoming financially autonomous and having increased interactions with other men, which also became increasingly common in the context of displacement, as women were relied on to help support the family. This led to controlling behaviors and physical, psychological, and economic IPV against women.

These findings provide important insights into the drivers of IPV in under-researched polygynous populations, and how displacement-related factors such as economic hardship and more equitable gender norms in host communities can impact these drivers. Although this study was conducted with Somali refugees in Ethiopia, our findings likely apply to other contexts in which displacement, and/or polygyny are common, and more research is needed to test if the pathways we identified hold true in these settings. Additionally, our findings have implications for IPV prevention programming more broadly. Such programming should more directly address jealousy and controlling behaviors in all types of relationships, and their link with IPV. Addressing conflict and relationship dynamics in polygynous households may require specialized content acknowledging the complex interactions and resource allocation between co-wives. Gender-transformative interventions that move away from masculinities that are built on the provider role and the introduction of alternative masculinities could be effective in reducing IPV against women, as they increasingly join the workforce around the world.

## Figures and Tables

**Figure 1 ijerph-19-05757-f001:**
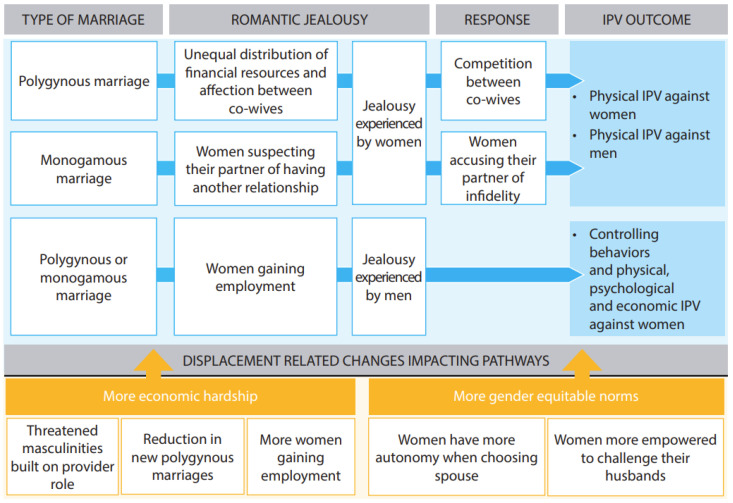
Identified pathways from romantic jealousy to intimate partner violence (IPV) against women and men in polygynous and monogamous relationships.

**Table 1 ijerph-19-05757-t001:** Study participant demographics.

Participant Demographics	Refugee Community Members	Elders/ Religious Leaders	Health Workers	UN/ NGO Workers	Community-Based Organization Workers	Policy Makers	Host Community Members	Total
	*n* (%)	*n* (%)	*n* (%)	*n* (%)	*n* (%)	*n* (%)	*n* (%)	*N* (%)
Nationality								
Somali	16 (100)	4 (100)	2 (100)	0 (0)	2 (100)	0 (0)	0 (0)	24 (80)
Ethiopian	0 (0)	0 (0)	0 (0)	2 (100)	0 (0)	2 (100)	2 (100)	6 (20)
Sex								
Women	8 (50)	0 (0)	0 (0)	2 (100)	1 (50)	1 (50)	1 (50)	13 (43)
Men	8 (50)	4 (100)	2 (100)	0 (0)	1 (50)	1 (50)	1 (50)	17 (57)
Mean, age (range in years)	31.7 (17–62)	61.3 (51–70)	45.5 (45–46)	22.5 (20–25)	36.5 (32–41)	31.0 (19–43)	40.5 (37–44)	36.8 (17–70)
Marital Status								
Single	5 (31)	0 (0)	0 (0)	2 (100)	0 (0)	0 (0)	0 (0)	7 (23)
Married	11 (69)	4 (100)	2 (100)	0 (0)	2 (100)	1 (50)	2 (100)	22 (73)
Separated	0 (0)	0 (0)	0 (0)	0 (0)	0 (0)	1 (50)	0 (0)	1 (3)
Mean length of time in camp (range in years)	6.8 (0.3–9)	7.6 (7–8)	7.5 (7–8)	1.7 (1–2.5)	8.0 (8)	8.0 (8)	N/A	7.3 (0.3–9)
Mean years of education (range in years)	3.6 (0–15)	2.0 (0–8)	15.0 (14–16)	10.0 (8–12)	10.0 (8–12)	7.5 (7–8)	2.5 (0–5)	5.6 (0–16)

## Data Availability

Not applicable.

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
