# Peer review of "Displacement, Polygyny, Romantic Jealousy, and Intimate Partner Violence: A Qualitative Study among Somali Refugees in Ethiopia"

_ijerph, 2022, doi:10.3390/ijerph19095757_

Round 1

Reviewer 1 Report

This study examined an important health issue. I have some suggestions for the authors to improve the manuscript.

  1. This study invited elders and religious leaders, organizational and service providers, policy makers and host community members as the members of groups for interview. The authors need to explain the reasons to invite them into this study as the sources of information.
  2. In the groups, husbands, wives and people outside the family may provide different information regarding their observation for the study issue. I would like to suggest the authors make some comparisons between.
  3. Polygyny exists not only in sub-Saharan Africa but also other countries or areas. The readers may expect to see more description regarding the role of polygyny in IPV under various culture or religious backgrounds.

Author Response

Point 1: This study invited elders and religious leaders, organizational and service providers, policy makers and host community members as the members of groups for interview. The authors need to explain the reasons to invite them into this study as the sources of information.

Response 1: The study was originally designed to inform the adaptation of an IPV prevention intervention for this context. The qualitative research included a purposive sample comprising women and men refugee community members, community elders, religious leaders, service providers and organization staff, as well as host community members in order to capture diverse views and perspectives on IPV and approaches to address IPV in this setting. This has been clarified on lines 124-125.

Point 2: In the groups, husbands, wives and people outside the family may provide different information regarding their observation for the study issue. I would like to suggest the authors make some comparisons between.

Response 2: We agree that making comparisons between information reported by different groups of participants would be imformative. While there were differences between participant groups on some topics discussed as part of the original study, the analysis for this paper which focused on perspectives related to romantic jealousy did not yield meaningful differences we can report on. This may be due to the relatively small sample size, and because jealousy was not discussed by all respondents. We have added text to the limitations section on lines 592-594.

Point 3: Polygyny exists not only in sub-Saharan Africa but also other countries or areas. The readers may expect to see more description regarding the role of polygyny in IPV under various culture or religious backgrounds.

Response 3: Thank you for this comment. Although a commonly cited driver of IPV, not much is known on the pathways through which polygyny is linked with IPV, which highlights the importance of this paper in addressing a gap in the literaure. Following your comment, we have added a more complete description of the literature available on polygyny and how it relates to relationship dynamics in the introduction on lines 63-74 as follows:

According to global Demographic and Health Surveys from 2000 to 2010, polygyny was reported to be legal or “generally accepted” in 33 countries around the world [1]. Polygyny has been traditionally present in cultures where the economic success of a family depends on the number of children available to work [2]. Research exploring the dynamics of polygyny have identified multiple sources of conflict between men and their wives, and among co-wives. One major source of conflict occurs when the family experiences economic hardship and having a large number of children in one household (as characteristic in polygynous unions) increases the financial burden for families [2]. Other identified sources of conflict within polygynous relationships include rivalries between co-wives and, in some cases between children of different mothers [3]. Polygyny has been widely associated with IPV [4], especially in sub-Saharan Africa [5], but little is known about the pathways through which polygyny leads to IPV.

References

  1. United Nations Department for Economic and Social Affairs, Population Division. Population Facts No. 2011/1. 2011. Available online: https://www.un.org/en/development/desa/population/publications/pdf/popfacts/PopFacts_2011-1.pdf (accessed on 28 April 2022).
  2. Elbedour, S.; Onwuegbuzie A.J.; Caridine, C.; Abu-Saad, H. The effect of polygamous marital structure on behavioral, emotional, and academic adjustment in children: a comprehensive review of the literature. Child Fam Psych. 2002, 5. 255-271. https://doi.org/10.1023/A:1020925123016
  3. Adegbite, O.B.; Ajuwon, A.J. Intimate partner violence among women of childbearing age in Alimosho LGA of Lagos State, Nigeria. Afr J Biomed Res. 2015, 18(1), 135-146.
  4. Coll, C.V.N; Ewerling, F.; García-Moreno, C.; Hellwig, F.; Barros, A.J.D. Intimate partner violence in 46 low-income and middle-income countries: an appraisal of the most vulnerable groups of women using national health surveys. BMJ Glob. Health 2020, 5, e002208. http://dx.doi.org/10.1136/ bmjgh-2019-002208
  5. Ahinkorah, B.O. Polygyny and intimate partner violence in sub-Saharan Africa: Evidence from 16 cross-sectional demographic and health surveys. SSM-Population Health2021, 13, 100729. https://doi.org/10.1016/j.ssmph.2021.100729

Reviewer 2 Report

Strengths: The proposed work investigates the phenomenon of jealousy in the intimate relationship from an original perspective, which considers situations of polygamy, and hypothesizes that jealousy may play a role in the expression of violence between partners.
It is well structured and has an adequate apparatus of references.

Limitations: only excerpts related to the interviews conducted appear in the presentation of the results. It would have been interesting, since the interviews were analyzed through content analysis, to report the occurrences of the thematic cores that emerged.

Author Response

Point 1: Strengths: The proposed work investigates the phenomenon of jealousy in the intimate relationship from an original perspective, which considers situations of polygamy, and hypothesizes that jealousy may play a role in the expression of violence between partners.
It is well structured and has an adequate apparatus of references.

Response 1: Thank you for this positive comment.

Point 2: Limitations: only excerpts related to the interviews conducted appear in the presentation of the results. It would have been interesting, since the interviews were analyzed through content analysis, to report the occurrences of the thematic cores that emerged.

Response 2: Thank you for this comment. The key themes that emerged from our analysis were described in the results as the causes of jealousy leading to IPV: (1) Unequal distribution of financial resources and affection between co-wives, (2) women suspecting their partner of having another relationship, and (3) male jealousy due to women gaining employment. The quotes included in the manuscript are examples of these themes. In response to this comment we have clarified that the pathways described throughout the results are built on the three main themes identified in our analysis (Lines 217-222).

Reviewer 3 Report

I have reviewed this article, which deals with the problem of intimate violence caused by romantic jealousies in the Ethiopian refugee camp of Bokolmayo, marked by social, cultural and religious factors, which makes intervention against violence necessary.

In the introduction, the manuscript highlights important gaps in the research about the pathways in which romantic jealousy can lead to different forms of intimate partner violence, mainly against women. However, the text concludes with no new pathways of those specified in the discussion, even though the authors have introduced the polygynous relationships and have explored the violence against men as well. How to strengthen the article regarding this?

The manuscript is presented as a second investigative phase, of qualitative interest, and based on data from other previous works. The passage from data collection to data analysis shows the selection of useful aspects of this qualitative research.

Among the strengths of this paper, I find it useful in suggesting which areas should be prioritized for interventions in the very limited space with such specific features. It is greatly appreciated that they suggest that the most appropriate gender-transformative interventions are those that move away from provider-based masculinities and introduce alternative masculinities. However, the conclusions do not end up providing new insights that can be used in a more universal perspective, taking into account the qualitative nature of the paper’s research.

Pointing out the limitations of the manuscript, the authors insist that this study focused on exploring the pathways from romantic jealousy to intimate partner violence in Bokolmayo. The text would be enriched if tables or figures that give a visual idea of the imaginary that is set in motion on the path from romantic jealousy to intimate partner violence were established. Somehow, through the selection of literal testimonies, they are pointed out, but they do not appear easily organized before the reader.
I also find it elegant and honest that the topic of romantic jealousy has not been induced in the questions (it appeared repeatedly in the answers, and it is the reason for this research). Jealousy in a community so marked by factors such as displacement, religion and polygamy has a dimension of power and privilege that could lead to even more universal conclusions than those expressed in the article, so I encourage the authors to complete it in that direction.
The article seems right in its ethical considerations, showing a profound respect for institutional procedures and especially for individuals in this sensitive area of life.

In my view, the article would gain greater visibility and public attention by having a shorter title.

As such a precise topic and specific field, there cannot be much previous specialized literature. In this sense, the references are accurate and adjusted to the issue.
In summary, the process is clear, the objectives are well defined, the article is well-structured and, as I said before, I would improve the visibility of the main aspects in a diagram, figure or table and I would complete the conclusions towards greater universality and wider audience.

Author Response

Point 1: I have reviewed this article, which deals with the problem of intimate violence caused by romantic jealousies in the Ethiopian refugee camp of Bokolmayo, marked by social, cultural and religious factors, which makes intervention against violence necessary.

Response 1: Thank you for reviewig this paper. 

Point 2: In the introduction, the manuscript highlights important gaps in the research about the pathways in which romantic jealousy can lead to different forms of intimate partner violence, mainly against women. However, the text concludes with no new pathways of those specified in the discussion, even though the authors have introduced the polygynous relationships and have explored the violence against men as well. How to strengthen the article regarding this?

Response 2: Through our analysis we identified three pathways from romantic jealousy to IPV against men and women in polygynous and monogamous marriages and these are described at the beinning of the discussion beginning on line 479. To further highlight the pathways we have added a figure on line 228 depicting them, and have refered back to this figure throughout the paper.

Point 3: The manuscript is presented as a second investigative phase, of qualitative interest, and based on data from other previous works. The passage from data collection to data analysis shows the selection of useful aspects of this qualitative research.

Response 3: Thank you for this positive comment.

Point 4: Among the strengths of this paper, I find it useful in suggesting which areas should be prioritized for interventions in the very limited space with such specific features. It is greatly appreciated that they suggest that the most appropriate gender-transformative interventions are those that move away from provider-based masculinities and introduce alternative masculinities. However, the conclusions do not end up providing new insights that can be used in a more universal perspective, taking into account the qualitative nature of the paper’s research.

Response 4: Thank you, we agree that the conclusions from this study could apply more universally to displaced and/or polygynous populations, although more research is needed to test this hypothesis. We have added this recommendation to our conclusion on lines 621-624. More broadly, the study findings, which highlight the importance of jealousy as a driver of IPV in all relationship types, are relevant to IPV prevention programming and suggest that program content should more directly address jealousy and controlling behaviors and their link with IPV. We have added this recommendation as well (lines 625-626).

Point 5: Pointing out the limitations of the manuscript, the authors insist that this study focused on exploring the pathways from romantic jealousy to intimate partner violence in Bokolmayo. The text would be enriched if tables or figures that give a visual idea of the imaginary that is set in motion on the path from romantic jealousy to intimate partner violence were established. Somehow, through the selection of literal testimonies, they are pointed out, but they do not appear easily organized before the reader.

Response 5: Thank you for this comment, we agree with the reviewer that a visual depicting the pathways would benefit the paper, and thus have created Figure 1 which has been added on line 228.

Point 6: I also find it elegant and honest that the topic of romantic jealousy has not been induced in the questions (it appeared repeatedly in the answers, and it is the reason for this research). Jealousy in a community so marked by factors such as displacement, religion and polygamy has a dimension of power and privilege that could lead to even more universal conclusions than those expressed in the article, so I encourage the authors to complete it in that direction.

Response 6: Thank you for this helpful suggestion. We have edited the beginning of our conclusion, removing the specific context in which this study was conducted (Bokolomayo refugee camp in Ethiopia), and suggesting that the evidence from this study could apply to polygynous and monogamous relationships more generally (lines 604-606). Later in the conclusion we also highlight that our findings provide important insights into the drivers of IPV in under-researched polygynous populations, and how displacement-related factors such as economic hardship and more equitable gender norms in host communities can impact these drivers. Although this study was conducted with Somali refugees in Ethiopia, our findings likely apply to other contexts in which displacement, and/or polygyny are common, and more research is needed to test if the pathways we identified hold true in these settings (lines 618-624).  

Point 7: The article seems right in its ethical considerations, showing a profound respect for institutional procedures and especially for individuals in this sensitive area of life.

Response 7: Thank you for this positive comment

Point 8: In my view, the article would gain greater visibility and public attention by having a shorter title.

Response 8: Thank you for this comment, in response we have shortened the title to “Displacement, polygyny, romantic jealousy and intimate partner violence: A qualitative study among Somali refugees in Ethiopia”

Point 9: As such a precise topic and specific field, there cannot be much previous specialized literature. In this sense, the references are accurate and adjusted to the issue.

Response 9: Thank you for this positive comment

Point 10: In summary, the process is clear, the objectives are well defined, the article is well-structured and, as I said before, I would improve the visibility of the main aspects in a diagram, figure or table and I would complete the conclusions towards greater universality and wider audience.

Response 10: Thank you. As noted above, in points 2 and 5 responses we have added a figure on line 228 and updated the conclusions towards greater universality on lines 604-626.